# Antifouling and Water Flux Enhancement in Polyethersulfone Ultrafiltration Membranes by Incorporating Water-Soluble Cationic Polymer of Poly [2-(Dimethyl amino) ethyl Methacrylate]

**DOI:** 10.3390/polym15132868

**Published:** 2023-06-29

**Authors:** Raja Muhammad Asif Khan, Nasir M. Ahmad, Habib Nasir, Azhar Mahmood, Mudassir Iqbal, Hussnain A. Janjua

**Affiliations:** 1Department of Chemistry, School of Natural Sciences (SNS), National University of Sciences and Technology, H-12, Islamabad 44000, Pakistan; 2Polymer Research Lab., Polymer and Composite Research Group, School of Chemical and Materials Engineering (SCME), National University of Sciences and Technology, H-12, Islamabad 44000, Pakistan; 3Atta-ur-Rahman School of Applied Biosciences (ASAB), National University of Sciences and Technology, H-12, Islamabad 44000, Pakistan

**Keywords:** antifouling, poly [2-(dimethyl amino) ethyl methacrylate] (PDMAEMA), hydrophilicity, porosity, polyethersulfone (PES) membranes, water flux

## Abstract

Novel ultrafiltration (UF) polymer membranes were prepared to enhance the antifouling features and filtration performance. Several ultrafiltration polymer membranes were prepared by incorporating different concentrations of water-soluble cationic poly [2-(dimethyl amino) ethyl methacrylate] (PDMAEMA) into a homogenous casting solution of polyethersulfone (PES). After adding PDMAEMA, the effects on morphology, hydrophilicity, thermal stability, mechanical strength, antifouling characteristics, and filtration performance of these altered blended membranes were investigated. It was observed that increasing the quantity of PDMAEMA in PES membranes in turn enhanced surface energy, hydrophilicity, and porosity of the membranes. These new modified PES membranes, after the addition of PDMAEMA, showed better filtration performance by having increased water flux and a higher flux recovery ratio (FRR%) when compared with neat PES membranes. For the PES/PDMAEMA membrane, pure water flux with 3.0 wt.% PDMAEMA and 0.2 MPa pressure was observed as (330.39 L·m^−2^·h^−1^), which is much higher than that of the neat PES membrane with the value of (163.158 L·m^−2^·h^−1^) under the same conditions. Furthermore, the inclusion of PDMAEMA enhanced the antifouling capabilities of PES membranes. The total fouling ratio (TFR) of the fabricated PES/PDMAEMA membranes with 3.0 wt.% PDMAEMA at 0.2 MPa applied pressure was 36 percent, compared to 64.9 percent for PES membranes.

## 1. Introduction

Polyethersulfone (PES) is a promising polymer in the field of advanced membranes technology that has recently been widely employed for a variety of biological and industrial applications [1,2]. One of the challenges in such applications is biofouling, which is detrimental to the performance of membranes including lower flux and separation. Biofouling is produced by the adherence and growth of microorganisms and the development of biofilms on the surface of the membrane [3], which is no doubt a serious problem in its long-lasting usage [4]. Fouling on the polymer membrane is generally because of the hydrophobicity of polyethersulfone, which makes the separation process uncertain and reduces its life owing to a greater energy requirement to drive water through the pores [5,6]. The characteristic adverse effects of biofouling in these kind of systems can be: (i) reduction in permeability of the membranes due to additional flux resistance sustained by biofilms, (ii) a decline in rejection rate due to the additional concentration polarization, (iii) higher energy consumption to tackle the increased pressure for biofilm resistance [7] decline in the flux [8], and (iv) a higher number of bacteria in the permeate. To reduce fouling, several modification procedures such as zwitterionic modification [9], hydrophilization [10,11], copolymer modification [12], and incorporating different nano particles such as TiO_2_ and SiO_2_ [13,14], have been used to increase membrane performance depending on a specific set of qualities for certain applications. Modifications using hydrophilic polymers and materials such as derivatives of poly (ethylene glycol) [15,16], zwitterionic polymers [17,18,19] or poly (vinyl pyrrolidone) (PVP) [20] are some of the often-used techniques to decrease fouling-related detrimental effects [21,22,23].

To achieve desirable characteristics such as hydrophilicity and antifouling performance of PES membranes, blending with hydrophilic polymers [24] has been the most effective and uncomplicated method. These blended filtration polymer membranes were fabricated by means of a cast solution of polyethersulfone poured in the solvents, such as 1-Methyl-2-pyrrolidinone (NMP) [25,26,27], dimethylacetamide (DMAc) [28], DMF [29], or a mixture of different kinds of solvents [30] together with some hydrophilic homopolymer, such as polyvinylpyrrolidone (PVP) [31] or polyethylene glycol (PEG) [32,33], as pore formers. This has led membranes with less susceptibility to fouling [34]. Dimethyl sulfoxide (DMSO) has been used as a casting solvent, and it also behaves as a permeability enhancer [35,36].

Among various hydrophilic polymers, poly [2-(dimethyl amino) ethyl methacrylate] (PDMAEMA)) is a renowned multifunctional polymer, which has been frequently utilized in biomaterials [37,38], double hydrophilic copolymers [39,40,41], and environmentally friendly coatings [42]. Because of the particular and uncommon chemical structure of PDMAEMA, it may respond to some external stimuli, such as pH [43], ionic strength [44], change in temperature [45] and many others exhibiting distinctive electrolytic effects and temperature responsiveness [46,47,48].References are updated

In the present study, we described the fabrication of new blended polymer membranes by the inclusion of PDMAEMA in the casting solution of PES. The primary objective of this research is to analyze the effect caused by the presence of PDMAEMA on the surface morphology, antifouling behavior, and filtration capabilities of PES/PDMAEMA-blended UF membranes. Thermogravimetric analysis (TGA) was used to examine the thermal stability and compatibility of PES/PDMAEMA-blended polymer membranes. Scanning electron microscopy (SEM), water contact angle measurements, and atomic force microscopy (AFM) were used to examine the impact of incorporating PDMAEMA on the morphology and hydrophilicity of blended polymer membranes. Moreover, newly fabricated polymer membranes were tested for their antifouling properties and filtration performance by filtrating bovine serum albumin (BSA) and water flux. A graphical overview of the present work is presented in Figure 1.

## 2. Materials and Methods

### 2.1. Materials

Polyethersulfone (Mol.Wt. = 58,000 g/mol) was obtained from BASF Company Ludwigshafen, Germany. 1-methyl-2-pyrrolidinone (NMP, 99%), polyvinyl pyrrolidone {(PVP), Mol.Wt. = 29,000 g/mol)}, dimethyl sulfoxide (DMSO, 99%), and ethyl acetate were acquired from Sigma-Aldrich (St. Louis, MO, USA). Poly (2-dimethylaminoethyl methacrylate) (PDMAEMA) was synthesized previously in the laboratory and used as received. Bovine serum albumin protein (BSA) (B005) was purchased from CAISSON Laboratories, Inc. (Smithfield, UT, USA). The chemicals used in this research were of analytical grade and did not require further purification.

### 2.2. Membrane Fabrication

The polymer membranes in this work were fabricated by phase inversion method, as reported previously, with some minor modifications [49,50]. Figure 2 represents the process of fabrication of membranes. PES powder was at first dried for 6 h at 50 °C. The PES cast solution, which contained 20 wt.% of PES in a solvent mixture of NMP and DMSO (4:1 *w*/*w*) and pore-forming polymer PVP (2% wt.%), was applied as a base polymer. Various concentrations of PDMAEMA were added in the mentioned solvent mixture, and then the mixture was thoroughly stirred for 3 h at 60 °C. Four types of ultrafiltration membranes were fabricated, which contained different concentrations of PDMAEMA, as shown in Table 1. The casting solution was then left for 24 h on magnetic stirring at room temperature, followed by degassing for 30 min. After homogeneous casting solution was formed, it was casted on polyester support using an Automatic film applicator (AFA-IV en-Moderner electromotor, Shanghai Modern Environment Engineering Technique Co., Ltd Shanghai, China). After applying the film, support was submerged in a casting tray filled with deionized water for 30 min. Twenty-four hours of room temperature air drying were performed on the solidified membranes produced on polyester support.

### 2.3. Characterization Techniques

To characterize the fabricated membranes, several characterization techniques and methodologies were used. The chemical structures of all the fabricated and modified polyethersulfone membranes were examined by FTIR (Nicolet 6700, M/s Thermo-Fisher Scientific, Madison, WI, USA) using OMNIC software (Version 7.1). This analysis was performed at resolution of 4 cm^−1^ within the specified range 4000–500 cm^−1^.

The membranes were characterized using scanning electron microscopy (SEM) (JEOL JSM 6490A, JEOL Ltd., Tokyo, Japan). Small portions of 1 cm^2^ each were cut from the fabricated membranes, gold coated, and placed on a steel stud with carbon tape to analyze morphology, cross-section, and topography. Atomic force microscopy (AFM) AC (JSPM-5200, JEOL Ltd., Tokyo, Japan) was utilized to measure the membranes’ surface roughness (Ra). Using the tapping method, 5 µm × 5 µm of an effective sampling area was achieved for the AFM study.

Thermogravimetric analysis (TGA) using (Q50, M/s TA Instruments, New Castle, USA) was conducted to evaluate thermal stability of membrane samples. Approximately 4~5 mg material was taken for each membrane to investigate the various phases of weight reduction (i.e., initial, degradation, and maximum temperature) by heating at 10 °C/min up to 700 °C in a N_2_ environment. (INSTRON-5966 Norwood, MA, USA). A universal testing machine was used to evaluate the mechanical characteristics of the membranes, such as their tensile strength and elongation, at 20 mm/min speed.

To measure the hydrophilicity of the samples of dried membrane samples, a sessile drop technique was used with the help of a custom-made equipment. On the surface of the fabricated membranes, an approximately 10 µL droplet of distilled water was discharged. Images were captured with a camera, to further process using ImageJ software (Version 1.51j). To reduce experimental error, the contact angle was determined using the static sessile drop technique, and an average of at least five measurements was obtained. The water retention capacity of a modified membrane was evaluated by immersing sample cuttings in H_2_O for 24 h. The weights of the dry membrane cuttings were determined after drying them in a vacuum oven for 12 h, and the water content (percentage) was estimated using Equation (1).
(1)$$\mathrm{Water}\;\mathrm{Intake}\left(\%\right)=\frac{\mathrm{Wet}\;\mathrm{Wt}.-\mathrm{Dry}\;\mathrm{Wt}.}{\mathrm{Wet}\;\mathrm{Wt}.}\times 100\;$$

A dry–wet weighing technique was adopted for measuring porosity of membranes and mean pore size [48,50]. After removing excess water using filter paper, the membrane samples were immersed in deionized water at 25 °C for 24 h and weighed. The wet membrane samples were air-dried in an oven for 24 h at 50 °C, and the corresponding dry weights were determined. The porosity (Ɛ) was computed as follows [50]:(2)$$\varepsilon =\frac{\left(\mathrm{Wet}\;\mathrm{Wt}.-\mathrm{Dry}\;\mathrm{wt}.\right)}{\rho_{W}A\delta }\times 100$$
where ρ_W_ represents the water density, which is (0.998 g·cm^−3^), A shows polymer membrane’s area, and δ is its thickness.

Prepared membranes were tested for water flux measurement in a dead-end UF-stirred filtration cell, which was fitted with a nitrogen gas cylinder. The inner diameter of the UF-stirred cell was 34 mm, with 300 mL volume capacity. Efficient filtration area was noted as 17 cm^2^. A nitrogen cylinder and pressure flowmeter served as pressure source and control system for the feeding flow. Before measuring water flux, each membrane sample was pre-compressed at a pressure of 0.35 MPa for about 30 min. Volumetric water-flux increased for all four samples as pressure increased from 0.1 to 0.35 MPa. Water flux of pure water through the membranes was tested by passing deionized water through the filtration assembly at different pressures like 0.1, 0.15, 0.2, 0.25, 0.3, and 0.35 MPa for 30 min. The following equation was used to calculate the permeate flux:(3)$$\mathrm{Jo}=\frac{\Delta V}{\mathrm{Am}\cdot \Delta t}\;$$
where Jo is membrane water flux, ΔV is the amount of water penetrated through the membrane (L), Am denotes the area of membrane (m^2^), and Δt is its permeation time(h).

### 2.4. Antifouling Ability of the Polymer Membranes

BSA was selected as the standard protein to analyze the antifouling ability of the fabricated polymer membrane. 1000 ppm (1 g/L) aqueous solution of BSA was prepared at room temperature. At 0.2 MPa, TMP, the flux of pure water J_w1_ (L·m^−2^·h^−1^) was initially observed. BSA solution was then filled in the flux-measuring cell to measure the flux of this solution J_b_. After filtrating solution containing BSA, the membranes were washed and rinsed with distilled water. The water flux of the pure water J_w2_ was calculated once more under conditions identical to those described in Figure 3.

The antifouling ability of the membranes was assessed by using the flux recovery ratio (FRR %) during filtration [9].
(4)$$\;\mathrm{FRR}\left(\;\%\right)=\frac{J_{w2}}{J_{w1}}\times 100$$

The fouling of the neat and modified membranes was then further evaluated with reference to the total fouling ratio (R_t_), reversible fouling ratio (R_r_), and irreversible fouling ratio, (R_irr_), which were computed by using these following equations [51].
(5)$$R_{t\;}\left(\%\right)=\left(1-\frac{J_{b}}{J_{w1}}\right)\times 100\;$$
(6)$$R_{r\;}\left(\%\right)=\left(\frac{J_{w2}-J_{b}}{J_{w1}}\right)\times 100$$
(7)$$R_{\mathrm{irr}\;}\left(\%\right)=\left(\frac{J_{w1}-J_{w2}}{J_{w1}}\right)\times 100$$

The following equation was used to calculate the BSA solution rejection percentage.
(8)$$R\;\%=\left(\;1-\frac{\mathrm{Cp}}{C_{f}}\;\right)\times 100\;$$
where C_p_ and C_f_ are, respectively, the permeate’s initial and final BSA concentrations (mg·mL^−1^)

## 3. Results and Discussion

### 3.1. FTIR Analysis of Membranes

Spectroscopic analysis of PES and PDMAEMA/PES blend membranes is presented in Figure 4, showing almost identical absorption peaks. These blended membranes exhibited the typical PES basic structure characteristics. The peaks at wavelengths 2821 cm^−1^ and 2922 cm^−1^ indicated the existence of -C–H stretching and the aromatic ring of the = C–H bond, respectively. Moreover, all the samples showed identical peaks at 1673 cm^−1^ and 1578 cm^−1^ representing C=C stretching vibrations of ester carbonyl group peaks [52]. Furthermore, peaks at 1322 cm^−1^ and 1240 cm^−1^ were linked with ether linkage between phenyl groups. PES via asymmetric vibrations of O=S=O asymmetric stretching bonds were shown by the presence of peaks at 1482 cm^−1^ [49,52].The peaks at 1151 and 1106 cm^−1^ have shown sulfone group presence in the PES base structure [1]. 

The distinctive absorption bands attributed to PDMAEMA emerged at 2922 cm^−1^ (C–H stretching of the –CH_3_ and –CH_2_ groups), 2821 and 2772 cm^−1^ (C–H stretching of the –N(CH_3_)_2_ group), and 1728 cm^−1^ (C=O group) when compared with pristine PES membrane [52], which showed that PDMAEMA chains were effectively present on the surfaces [53]. The symmetric stretching vibration of the C–N observed at absorption band 1150 cm^−1^ is associated with the presence of DMAEMA [54]. 

### 3.2. Surface Energy and Hydrophilicity

Hydrophilicity is a substantial element in determining the antifouling characteristics. The hydrophilicity of the surface was assessed in this work by applying the sessile drop technique to determine the contact angle of the water. It is widely assumed that lesser the contact angle, the greater the hydrophilicity will be [15].

Hydrophilicity increased with the increasing concentration of PDMAEMA in all modified composite membranes. MVD0, which is a pristine membrane without any additive, exhibited the highest contact angle of 60.5 °C, and blending with PDMAEMA significantly lowered the contact angles of MVD1, MVD2, and MVD3 to 50.4 °C, 44.2 °C, and 37.1 °C, respectively, depending on the concentration of PDMAEMA. The contact angle dropped when the amount of PDMAEMA in the cast solution was raised, indicating that when more hydrophilic additive is added to the polymeric matrix, the surface becomes more hydrophilic, resulting in a lower contact angle. Membranes with lesser contact angles than the pristine PES membrane exhibited better hydrophilicity after blending modification. It is because of the protonation of the PDMAEMA molecule that PDMAEMA acts as a weak polybase below its pKa value of ~7.4 and becomes protonated—thus, cationic—whereas above this pH, it is deprotonated and neutral. Because of its pH and temperature-sensitive properties, PDMAEMA is being employed for a range of applications in biotechnology, emulsions, and drug delivery, etc. [55]. 

The surface energy of the blended membranes was estimated by adding PDMAEMA, as given in Table 2. The pristine PES membrane exhibited the surface energy at 53.72 J/m^2^. As the roughness of these modified polymer membrane surfaces increased and the contact angle decreased, the surface energy increased for MVD1, MVD2, and MVD3 at 58.94, 61.80, and 64.71 J/m^2^, respectively. Figure 5 presents the increase in surface energy caused by decreasing the contact angle. These values are affected by the physio-chemical interaction with the water molecule. Equation (9) was used to calculate the apparent surface energy by combining the equilibrium contact angle, Chibowski correlation, and the Young equation [56].
(9)$$\gamma_{s}=\frac{\gamma l}{2}\;\left(\;1+\cos\theta_{\mathrm{Eq}}\;\right)\;$$
where 𝛾_s_ is the apparent surface energy, 𝛾𝑙 is surface tension of liquid, and 𝜃_Eq_ is the contact angle.

### 3.3. Membrane Porosity and Wettability

Membrane porosity and water intake are mutually dependent variables, since increasing the porosity of the membrane will in turn increase water intake and vice versa [49]. Table 2 depicts water intake and membrane porosity. According to the porosity and water intake analysis, the pristine PES(MVD0) membrane has an intermediate value of 42.86% and 32.43%, respectively.

The MVD1 sample has shown higher porosity (53.94%) values, indicating more water intake (37.80%). Similarly, by increasing the proportion of hydrophilic polymer, porosity in MVD2 and MVD3 also increased to 63.23% and 68.62%, respectively. Water intake was also observed for both the MVD2 and MVD3 membranes at 40.15% and 42.45%, respectively. Due to the relatively higher number of pores, it was noticed that the water intake and porosity of the membranes with a higher concentration of polymer were greater than other membrane samples. Figure 6 shows the results of the porosity measurements, which shows that by adding PDMAEMA, the porosity of the membrane was found to be increasing. Pore formation would be facilitated by the presence of hydrophilic PDMAEMA. Therefore, increasing the concentration of PDMAEMA in the membranes causes the development of higher number of pores, which leads to an increase in porosity [49]. According to these observations, the presence of additives enhances the porosity of the membranes in general. It is evident that as the additives are increased, porosity rises due to the development of larger finger-like cavities [28].

### 3.4. Thermal Stability

TGA was used to examine the thermal stability of polyethersulfone and (PES/PDMAEMA) membranes. The thermal degradation curve of the membrane samples from ambient temperature to 700 °C is displayed in Figure 7. The main degradation took place in a single phase. The maximum single weight losses were found in the temperature range between 350 and 520 °C. The evaporation of water molecules and other volatiles causes a minor degree of weight loss in all four membranes between 100 °C and 150 °C [1,9]. The substantial weight loss of pure PES starts around 445.97 °C, showing that PES has high thermal stability [57].

The onset temperature representing the thermal degradation (Ti) of MVD1, MVD2, and MVD3 was observed at 367.81 °C, 386.04 °C, and 389.80 °C, respectively. MVD1 showed the lowest decomposition temperature of the three membranes, which might be attributed to asymmetrical behavior with the addition of PDMAEMA in the PES pristine membrane, subsequently making it less stable. The temperature at the maximum degradation (Tmax) of all membranes ranged between 367.81 °C and 445.97 °C, owing largely to their thermal degradation [1]. When the PDMAEMA concentration increased in the casting solutions, the T_max_ also increased to some extent from 367.81 °C in MVD1 to 389.80 °C in MVD3, but in general the thermal solidity of the membranes after adding PDMAEMA was observed as lower compared to the neat PES membrane. The results indicate that adding PDMAEMA to the membranes lowered their thermal stability marginally, which relates to oxidation, volatile degradation, thermal decomposition of side groups, and also initiating the main chain disintegration [57].

### 3.5. Mechanical Testing

Two crucial factors that determine the mechanical stability of a membrane are tensile strength and elongation at break. Table 3 summarizes the mechanical properties of PES polymer membranes that have been modified by adding PDMAEMA. It is obvious that as the polymer concentration increases, tensile strength and modulus also increases, as did the elongation at the break. However, comparing tensile strength data from various studies is complex, since the mechanical testing of the different membranes is reliant on the specific area of the material with its exact measurement.

The inclusion of the hydrophilic polymer into the PES polymer medium can enhance their tensile strength [58,59]. The tensile strength of the pristine PES membrane was recorded at 10.322 MPa, whereas the elongation of the membrane at break was 37.56%. These properties of the membranes increased when the percentage composition of PDMAEMA increased, as indicated in Table 3. The tensile strengths of the modified polymer membranes were observed at 11.519 MPa, 13.202 MPa, and 15.527 MPa for MVD1, MVD2, and MVD3, respectively. Higher tensile strength and strain suggests that the membrane is neither brittle nor ductile, nor readily broken or damaged while being subjected to a greater workload. These factors influence the duration of the membrane lifetime [58].

Similarly, elongation at break of the polymer membranes also enhanced with increasing PDMAEMA concentration, that is, 27.76%, 36.44%, and 38.16% for MVD1, MVD2, and MVD3 membranes, respectively, as shown in Figure 8. The increase in mechanical properties is attributed to PDMAEMA, which is well-known for its mechanical strength, as well as the better ability of dispersion of PDMAEMA in the membrane solution, which results in a solid contact between PDMAEMA and the polymer matrix. Furthermore, the presence of PDMAEMA inhibits the development of macrovoids, enhancing the mechanical characteristics of membranes. It also demonstrates that MVD3 composite membranes have the highest tensile strength of the other modified membranes. Also, the crosslinking density rises to a considerable level as the content of DMAEMA increases, perhaps improving tensile strength [59].

### 3.6. Morphological Analysis by SEM 

The membrane’s surface and cross-sectional morphology were evaluated by SEM, as seen in Figure 8. It is obvious that when PES and PDMAEMA are blended, the surface morphology of the membranes changes noticeably. The top surface of the PES membrane sample exhibits pores of varying sizes, and it is evident that increasing the amount of polymer concentration increased the porosity throughout the surface. As the weight percentage of PDMAEMA increases, a higher number of pores seems to develop [34]. 

Figure 9 shows cross-sections and surface images of the membranes. The asymmetrical structures of membranes are depicted in these images, which also show dense top layers with porous sublayers and macro voids on the lower ends [28]. Morphology of the membranes changed by adding PDMAEMA. The porous behavior of the altered polymer membranes increased by adding PDMAEMA and was highest for the MVD3 membrane, but the pure membrane MVD0, which did not include any PDMAEMA, had a smaller number of pores. The thermo-dynamic stability of the polymer blend system was expected to decrease with the addition of PDMAEMA. This factor causes de-mixing of both the solvent and non-solvent, which allows an increased membrane porosity [49]. By increasing polymer percentage, the membrane’s morphology develops through a series of channel-like holes, followed by finger-like pores, and lastly a sponge-like structure, as shown in Figure 9. As it is obvious in Figure 9d that the number of channels and pores are significantly higher than that of Figure 9b, this clearly indicates that the interconnectivity increases with the addition of PDMAEMA.

### 3.7. Surface Examination by AFM

Figure 10 presents three-dimensional figures of the fabricated membranes obtained by using AFM. Morphological changes over the surface of membrane can be imputed by adding PDMAEMA to the membranes. The surface roughness of the neat MVD0 membrane was found at a low level relative to MVD1, MVD2, and MVD3 and was noticed to increase with the addition of PDMAEMA. The parameters of the surface roughness that were observed for these membranes, given in Table 2, are expressed as mean roughness (Ra).

### 3.8. Flux Studies

The ultrafiltration characteristics of the pristine and improved membranes were assessed by determining the water flux at different pressures, as shown in Figure 11. With increasing pressure, all these membranes exhibited an increase in volumetric water flux. Figure 11 illustrates the determined permeate flow of the membranes as a function of pressure. The pressure applied was directly proportional to the water flux of all the membranes. As the applied pressure increased, the water flux of these membranes also increased. This is because an increase in transmembrane pressure increases the driving power for pure water permeation across membranes [49].

The neat PES membrane with the least values of water flux (55.06 L·m^−2^·h^−1)^ at minimum pressure of 0.05 MPa was observed. With the increase in constant pressure, the permeation also increased and was observed as 619.50 L·m^−2^·h^−1^ at 0.35 MPa. Similarly, water flux of the modified membranes also showed a significant effect with the change in applied pressure as well as the concentration of the additives. The water flux of PES/PDMAEMA membranes with 3.0 wt.% drastically increased to 861.90 L·m^−2^·h^−1^ at 0.35 MPa due to the increased porosity. Figure 11 illustrates the effects of the addition of PDMAEMA PES modifiers on water flux at different pressures.

When associated with the pristine PES membrane, the water flux seemed to improve in the modified membranes. At an applied pressure of 0.35 MPa and PDMAEMA with 1.0 wt.%, the flux of pure water reached a peak of approximately 649.96 L·m^−2^·h^−1^. Similarly, when the polymer concentration raised up to 2.0 wt.%, the water flux also enhanced to 734.21 L·m^−2^·h^−1^ at the same pressure. This rise in the water flux is due to the higher porosity of the membranes with different percentages of the polymer. The improvement in membrane hydrophilicity with PDMAEMA also facilitated water diffusion through these modified membranes. The morphology of membranes also has an impact on water flux. The internal configuration of the membranes altered from relatively closed to highly interconnected pores, as revealed in the SEM images of the membranes represented in Figure 9, which resulted in improved water flux. 

The flux rate of the modified membranes is significantly higher than the pristine and keeps on increasing with respect to the PDMAEMA concentration. These results are in accordance with the previous reports in which somewhat similar hydrophilic polymers were embedded into the PES membrane [60,61]. Figure 11b shows a comparative behavior between the flux of pure water and BSA solution. BSA is widely known for its sticky characteristics, which is why it is generally used as a model protein [15]. As compared to pure water, the flux rate of BSA at 1000 ppm is much lower [51]. For example, at a particular pressure of 0.2 MPa, the pure water flux was noted as 163.16 L·m^−2^·h^−1^, whereas the BSA solution flux was recorded at 57.26 L·m^−2^·h^−1^. Similarly, increasing the amount of PDMAEMA in both pure water and BSA solution showed a noticeable enhancement in the flux rate. As for the MVD3 membrane having 3 wt.% PDMAEMA, the flux for pure water increased up to 330.40 L·m^−2^·h^−1^ as opposed to 211.45 L·m^−2^·h^−1^ for the BSA solution. All the fabricated membranes showed a rapid flux decline while filtrating BSA solutions when compared with the pure water from the membranes. This variation may be because of the deposition of BSA on the membrane surface by blocking the pores and channels [62]. This observed change in the flux rate can be due to the higher porosity and complex structural feature of both PDMAEMA and BSA protein that promote their permeation through the membranes under pressure.

Figure 12 depicts the percentage of BSA rejection by the membranes. Except for the neat PES membrane, the rejection characteristics of other modified membranes were almost similar. It was analyzed that the PDMAEMA-added membranes presented a higher BSA rejection than the pure PES membranes. The BSA rate of rejection by the pure PES membrane (MVD0) was about 87%, while the rejection rate of all membranes added with PDMAEMA was almost 98%. This is because the surfaces of the PES/PDMAEMA membranes had a relatively better hydrophilic nature. This result might be due to improved surface hydrophilicity, larger pore sizes, and improved membrane structure [51]. One of the possible reasons may be the presence of carboxyl groups, having interactions between water molecules and carboxyl groups, as well as the protonation of the hydrophilic polymer.

### 3.9. Antifouling Studies

The flux recovery rate is another significant parameter to determine antifouling characteristics [62]. A better flux recovery rate often denotes a stronger antifouling performance [9]. To assess the membrane antifouling properties, each membrane’s time-dependent flux was measured by employing BSA protein as a typical foulant. Because of concentration polarization and protein contamination, the permeation flux of the membranes declines quickly from pure water to BSA solution, and the flux of BSA-filtered membranes was much lower than the flux of pure water, as illustrated in Figure 11b. Proteins are widely recognized for their capacity to accumulate on the surface of membrane and block the channels by entrapping themselves in pores, all of which negatively effects the permeability characteristics of the membrane.

The membranes underwent thorough cleaning after filtrating the BSA solution, and to evaluate water flux once again. The increase in the water flux was seen as indicating the maximum removal of the BSA molecules from the membrane surface after the washing process. This suggests a better flux recovery. The flux recovery ratio (FRR%) for each membrane was determined based on the flux values obtained to estimate the membrane’s antifouling capacity. The flux recovery ratio of the membranes is basically a measure of their antifouling characteristics. The antifouling performance improves as the FRR value increases [9]. MVD0 has the least FRR value, which is about 75.9%, whereas the FRR value has increased as the percentage of PDMAEMA increased in the PES membranes, as shown in Figure 13a. The FRR value was observed as 83.8%, 86.5%, and 90.0% for MVD1, MVD2, and MVD3, respectively. A higher FRR value also suggested that the higher number of adsorbed BSA particles were eliminated from the surface of membranes during the washing process.

Figure 13b depicts the overall membranes contamination in terms of total (Rt), reversible (Rr), and irreversible fouling (Rirr). All PES membranes incorporated with PDMAEMA had lower Rr and Rirr than pure polyethersulfone membranes. Similarly, Figure 13c indicates the fraction of the flux reduction, which is irreversible. 

The pristine polyethersulfone membrane (MVD0) demonstrated the greatest Rt value, indicating that pure PES membranes are more prone to fouling. The Rt value for MVD0 was observed as 64.9% and decreased with the increase in FRR and percentage of the polymer. The Rt value for MVD1, MVD2, and MVD3 was found to be 54.1%, 44.9%, and 36%, respectively. Both the reversible and irreversible fouling ratios reduce as the amount of PDMAEMA increases, with the MVD1 having the maximum and MVD3 membrane having the lowest.

## 4. Conclusions

In this research, neat polyethersulfone (PES) and modified PES membranes by incorporating cationic poly [2-(dimethyl amino) ethyl methacrylate] (PDMAEMA) were formulated by the phase inversion method. Various membrane characteristics such as morphology, filtration performance, and antifouling characteristics were affected by the addition of PDMAEMA in the polymer casting solution. It was observed that PES membranes modified with PDMAEMA contain larger finger-like structures as compared to pristine membranes. The pores interconnectivity between the bottom layers and sublayers increased. The hydrophilicity of the membranes was improved by the addition of the hydrophilic polymer. PDMAEMA and DMSO both exhibit polar characteristics, which enhance their affinity towards water (polar) molecules and thus results in the higher hydrophilicity of the fabricated membranes. The surface energy and water contact angle data explain their effect on the hydrophilicity and morphology of the modified membranes. Moreover, the membrane porosity was also raised from 53.94% to 68.62% as 3 wt.% PDMAEMA was poured into the casting solution. The inclusion of the hydrophilic polymer increased the modified membranes’ water flux as well. The flux of pure water for the membrane with 3.0 wt.% loading was found at a maximum of 330.4 L·m^−2^·h^−1^ at an applied pressure of 0.2 MPa. Moreover, it was examined that increase in the pressure in turn increased the water flux of the membranes. The flux of (MVD3) with 3.0 wt.% PDMAEMA increased from 127.48 L·m^−2^·h^−1^ to 861.9 L·m^−2^·h^−1^ at pressures 0.05 MPa and 0.35 MPa, respectively. Furthermore, the antifouling nature of the modified membranes were also improved. PES/PDMAEMA membranes had a greater flux recovery ratio and smaller Rt values. The Rt was decreased while the FRR values were raised when the quantity of PDMAEMA in the PES membranes was increased.

## Figures and Tables

**Figure 1 polymers-15-02868-f001:**
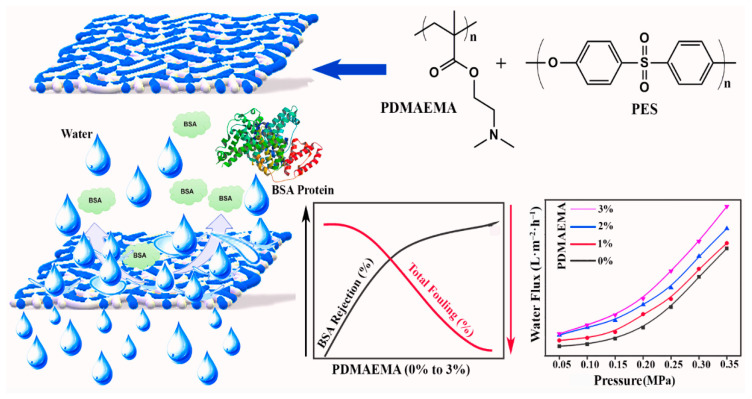
Graphical overview to present the effect of PDMAEMA on fouling and water flux of the PES membrane.

**Figure 2 polymers-15-02868-f002:**
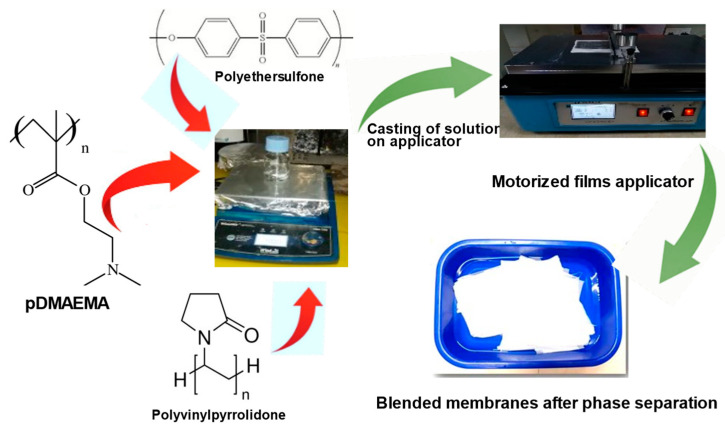
PES and PES/PDMAEMA membranes preparation.

**Figure 3 polymers-15-02868-f003:**
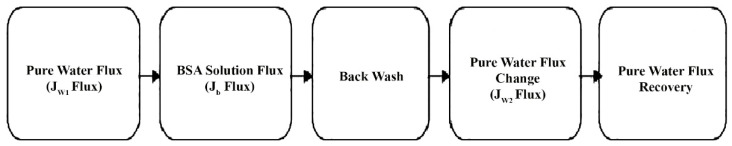
Flow chart for representing membrane fouling and membrane flux recovery.

**Figure 4 polymers-15-02868-f004:**
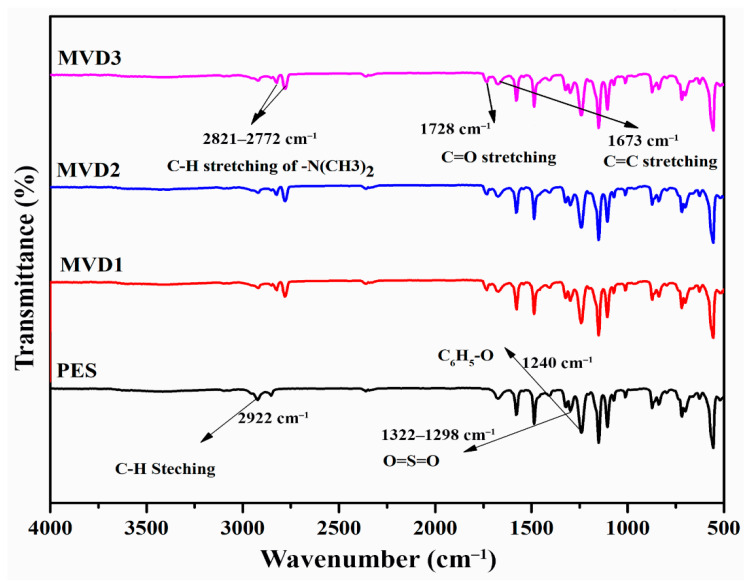
FTIR spectra of pristine PES and modified membranes incorporated with PDMAEMA.

**Figure 5 polymers-15-02868-f005:**
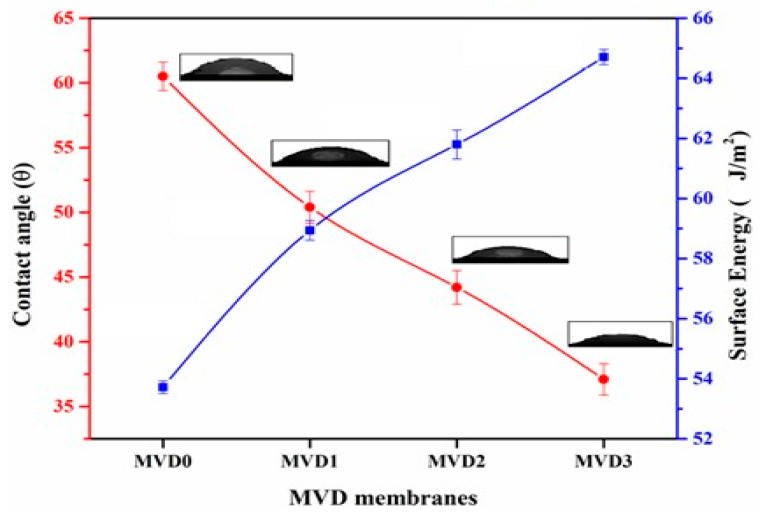
Contact angles and surface energies of the PES and modified PDMAEMA membranes.

**Figure 6 polymers-15-02868-f006:**
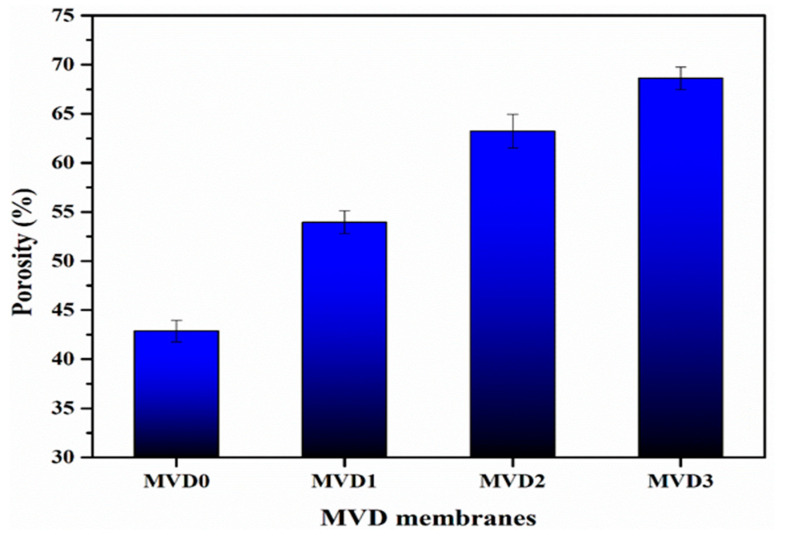
The porosity measurement of the PES and modified PDMAEMA membranes with their codes mentioned in Table 1.

**Figure 7 polymers-15-02868-f007:**
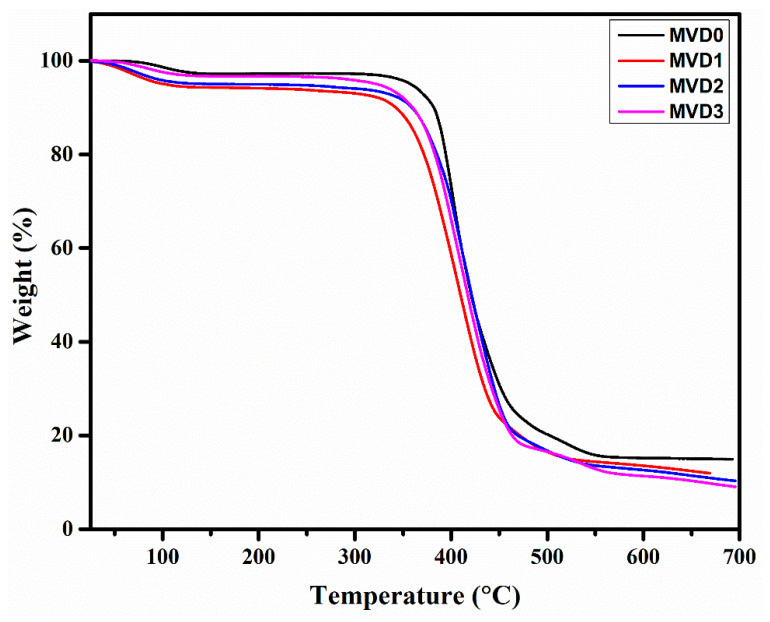
Thermal degradation studies of the PES and PES/PDMAEMA membranes.

**Figure 8 polymers-15-02868-f008:**
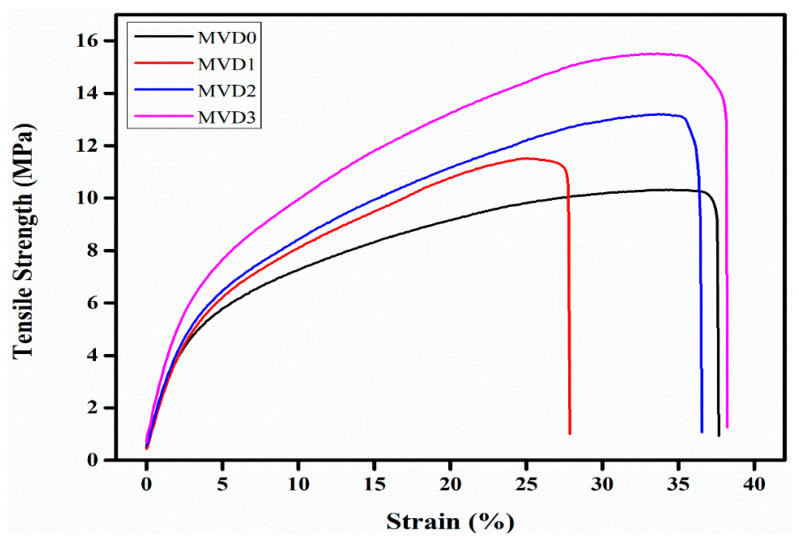
Tensile stress–strain curves for PES modified membranes of MVD0, MVD1, MVD2, and MVD3 having 0, 1, 2, and 3% PDMAEMA.

**Figure 9 polymers-15-02868-f009:**
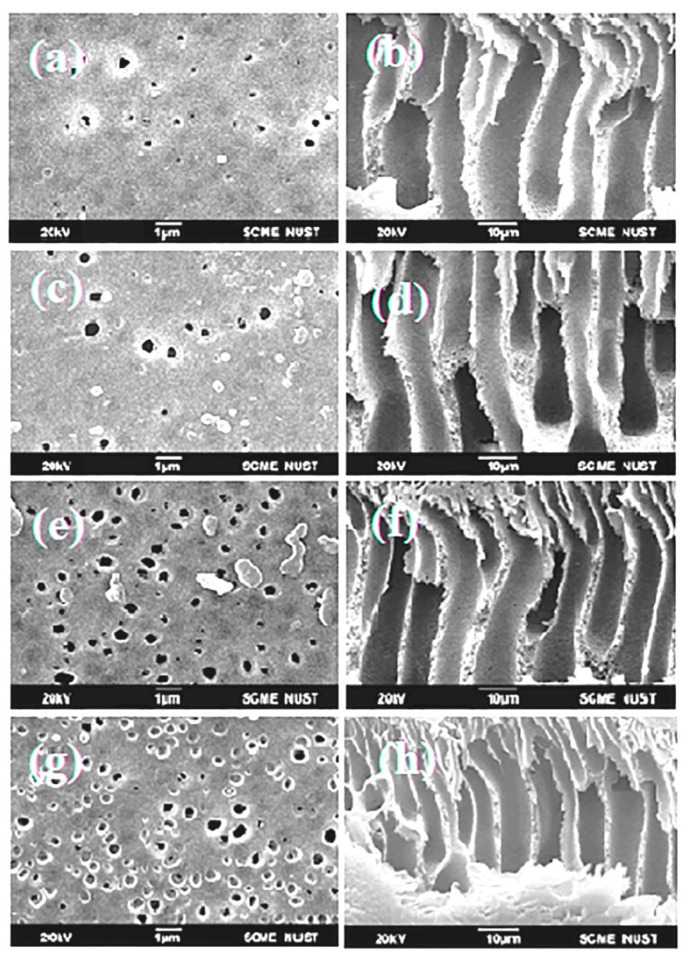
SEM images with cross-sectional and surface morphology of (**a**,**b**) MVD0, (**c**,**d**) MVD1, (**e**,**f**) MVD2, and (**g**,**h**) MVD3 membranes.

**Figure 10 polymers-15-02868-f010:**
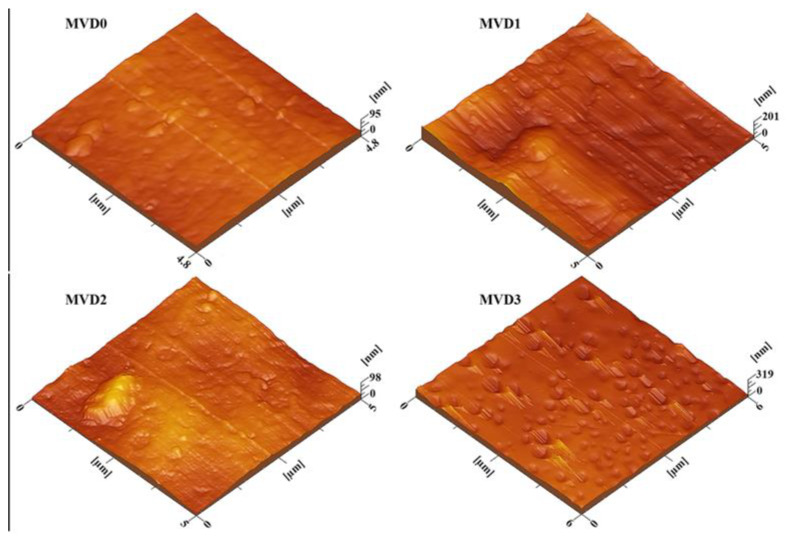
Three-dimensional AFM images of pristine PES membrane (MVD0), PES with 1% PDMAEMA (MVD1), PES with 2% PDMAEMA (MVD2), and PES with 3% PDMAEMA (MVD3).

**Figure 11 polymers-15-02868-f011:**
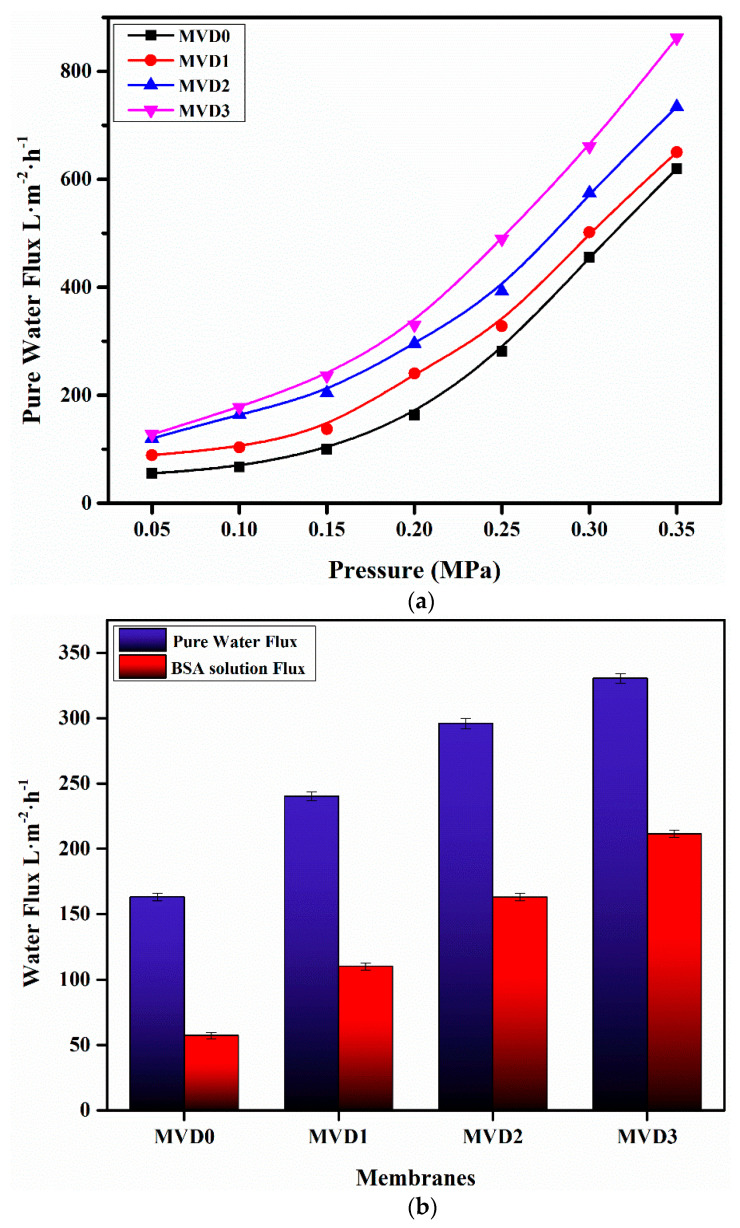
Studies of the ultrafiltration abilities of the pristine and improved membranes incorporated with PDMAEMA: (**a**) Pure water flux at elevated pressure; (**b**) water and BSA solution flux rate at 0.2 MPa applied pressure.

**Figure 12 polymers-15-02868-f012:**
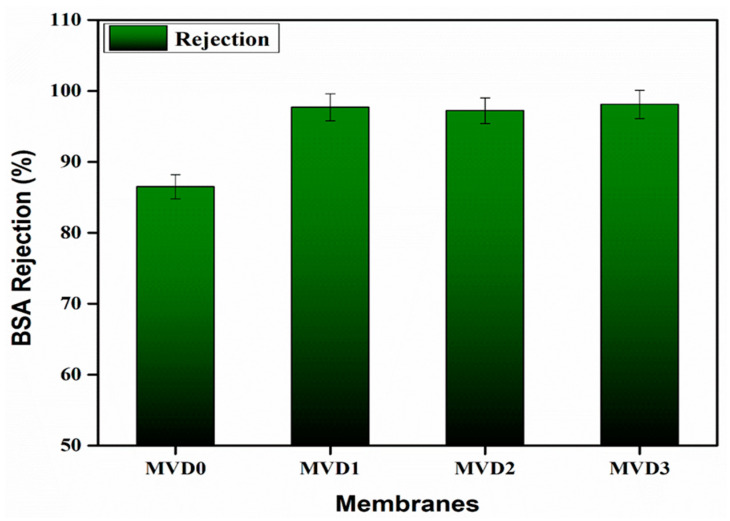
BSA rejection percentage of the membranes.

**Figure 13 polymers-15-02868-f013:**
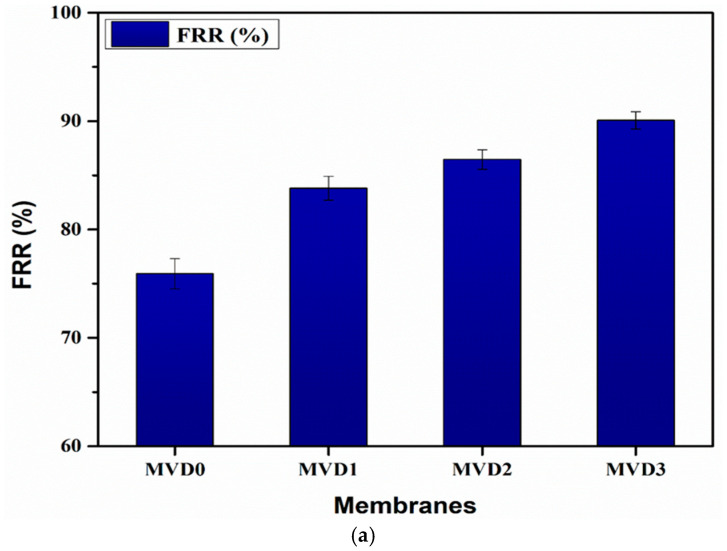
Antifouling behavior: (**a**) flux recovery ratio of the membranes; (**b**) fouling of the membranes in terms of total (Rt), reversible (Rr), and irreversible fouling (Rirr); (**c**) irreversible flux reduction (Rirr/Rt).

**Table 1 polymers-15-02868-t001:** Materials and compositions of the casted polymer flat sheet membranes.

Membrane Code	Composition as a Percentage (wt.%)				
	PES	PVP	PDMAEMA	NMP	DMSO
MVD0	20	2	-	62.4	15.6
MVD1	20	2	1	61.6	15.4
MVD2	20	2	2	60.8	15.2
MVD3	20	2	3	60	15

**Table 2 polymers-15-02868-t002:** Contact angle, surface energies, water intake, and porosity readings.

Sample Code	Contact Angle θ (°)	Surface Energy (J/m^2^)	Water Intake (%)	Porosity (%)	Ra (nm)
MVD0	60.5 ± 1.10	53.72 ± 0.21	32.43 ± 1.33	42.86 ± 1.10	11.0
MVD1	50.4 ± 1.23	58.94 ± 0.33	37.80 ± 1.16	53.94 ± 1.15	11.1
MVD2	44.2 ± 1.30	61.80 ± 0.48	40.15 ± 1.61	63.23 ± 1.70	18.1
MVD3	37.1 ± 1.21	64.71 ± 0.25	42.45 ± 1.54	68.62 ± 1.14	22.1

**Table 3 polymers-15-02868-t003:** Mechanical analysis results of membranes.

Sample Code	Stress at Max. Load (MPa)	Strain at Break (%)	Tensile Modulus (MPa)
MVD0	10.322	37.56	89.43
MVD1	11.519	27.76	125.34
MVD2	13.202	36.44	105.63
MVD3	15.527	38.16	121.09

## Data Availability

All the data is provided in the manuscript and is available for the readers.

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
