# Peer review of "Antifouling and Water Flux Enhancement in Polyethersulfone Ultrafiltration Membranes by Incorporating Water-Soluble Cationic Polymer of Poly [2-(Dimethyl amino) ethyl Methacrylate]"

_polymers, 2023, doi:10.3390/polym15132868_

Round 1

Reviewer 1 Report

The authors report an experimental study of ultrafiltration (UF) membranes prepared from blends of poly(2-dimethyl amino ethyl methacrylate) (PDMAEMA) with polyethersulfone (PES). The addition of PDMAEMA changes membrane properties especially hydrophilicity and surface morphology which in turn affect filtration and fouling performance.

The authors should address the following concerns prior to publication.

1.           General comment. The manuscript would benefit from careful review for spelling and grammar.

2.           General comment. The authors examine the effect of adding PDMAEMA and DMSO to the membrane casting solution on membrane properties and performance. The results are compared to a membrane prepared from only PES at a different polymer concentration. What is the effect of PES concentration on the membrane? It would seem appropriate to compare membranes prepared using the same total polymer composition. Additionally, how does viscosity vary between the different polymer solutions used? This will have a dramatic impact on the results if formation conditions (temperature and time) are held constant.

3.           Page 3. Please disclose the source of the polyester support and its nominal characteristics such as porosity, pore size, and thickness.

4.           Page 3. The polyester support was coated with the polymer solution. What was the cast film thickness? How was the value used determined? What was the time interval between casting and immersion in the deionized water bath?

5.           Page 3. How did the authors verify that their solutions of PES and PDMAEMA were homogeneous?

6.           Page 3. The authors state the PES only solution contains “20 wt. % of PES in solvent mixture of NMP and DMSO (4:1 w/w) and pore forming polymer PVP (2% wt. %).” However, Table 1 indicates MVD0 does not include DMSO. Please clarify. Additionally, if DMSO was not used to form MVD0, what impact will this have on the comparisons with MVD1, MVD2, and MVD3?

7.           Page 5. How did the authors determine 12 hours drying in a vacuum oven was sufficient to dewater the samples?

8.           Page 6. In Figure 3, please clarify how the pure water flux change was calculated. The figure appears to refer to Jw1 as the change.

9.           Page 5-6. The discussion of contact angle results is repetitive. This section should be rewritten to remove duplicate statements of 1) contact angle values, 2) effect of PDMAEMA on hydrophilicity, and 3) effect of PDMAEMA on contact angle.

10.         Page 10. The authors state MVD1 starts decomposition at the lowest temperature. Did the authors dry the samples before TGA testing? What is the difference in water content of the starting samples? Were the results influenced by residual water?

11.         Page 11. The authors report stress at max load to 5 significant figures. Are their measurements that accurate? What is the error in the measurement? What is the sample to sample variation?

12.         Page 13. The images in Figure 10 should be improved. In particular, the z axis should be expanded to better visualize differences between the membranes.

13.         Page 14. Figure 11 indicates the relationship between pure water flux and pressure is not linear. What is the source of the non-linearity?

14.         Page 14. Figure 11 indicates water fluxes increase with increasing PDMAEMA content. If flux is scaled by porosity, do the curves collapse onto one line? Plotting a scaled flux (scaled flux = flux/porosity) versus pressure would address this.

15.         Page 14. The authors state water flux might increase in the PDMAEMA membranes due to increased water diffusion resulting from increased hydrophilicity. Diffusive water transport rates through the solid membrane are expected to be orders of magnitude smaller than convective water transport rates through pores. Please justify this statement.

16.         Page 15. The discussion of why MVD0 has lower BSA rejection compared to the PDMAEMA containing membranes is not adequate. Figure 9 suggests MVD0 has the lowest porosity and smallest pores. Why would MVD0 have lower rejection? Moreover, one would expect BSA fouling to further reduce pore size and thereby increase rejection with time.

17.         Page 16. The authors state that the increase in water flux after cleaning the BSA fouled membranes indicates “complete removal of BSA molecules from membrane surface after washing process”. How did the authors confirm complete removal? The pure water flux does not return to its original value which seems to indicate some BSA is not removed and blocks membrane pores.

18.         Page 16. The authors refer to Rirr but Equation (7) uses the notation Rir. Consistent notation should be used.

19.         Page 17. It would be interesting to plot (Rirr/Rt) in Figure 13 to indicate the fraction of the flux reduction that is irreversible.

20.         Page 17. Conclusion. The authors state pore interconnectivity increases with PDMAEMA addition. This is not clear from Figure 9. Additionally, I do not believe this is discussed elsewhere in the manuscript.

21.         Page 17. Conclusion. The authors restate the quantitative results they report in the manuscript. However, they do not discuss why the changes occurred with addition of PDMAEMA and DMSO. A discussion of the physics and chemistry that led to the changes in membrane properties and performance should be added.

Author Response

Respected Reviewers'

We would like to express our appreciation for your constructive comments, which helped us a lot to improve our paper. We are grateful for your valuable suggestions.

Reviewer 2 Report

This paper has presented a method to improve the anti-fouling performance and water flux of PES ultrafiltration membrane by blending hydrophilic polymer. It is an interesting work. However, before it is accepted, this manuscript should be improved and some suggestion were listed as below.

1.       The keywords are not inappropriate. “Poly [2- (Dimethyl amino) ethyl Methacrylate]” should be included.

2.       An abbreviation appears for the first time, the full name must be written first. For example, “Polyethersulfone (PES)”.

3.       Figure 1 could not describe the membrane formation process and the role of PDMAEMA. And the horizontal axis in the lower right corner figure would be wrong.

4.       In Table 1, why the ratio of NMP/DMSO of MVD0 was not 4:1 as other membranes?

5.       The polymer concentration of these 4 membranes were same as 20 wt.%. Why the porosity of these 4 membranes were so different? Please explain the reason.

6.       Please compare the performance of the obtained membrane in this paper with that in the latest reports.

7.       Molecular cut-off data of these 4 membranes should be added.

8.       Only physical cleaning was done in this paper. Chemical cleaning should be added.

9.       Long-term operation test should be added.

Author Response

Respected Reviewer

We would like to express our appreciation for your constructive comments, which helped us a lot to improve our paper. We are grateful for your valuable suggestions.

Round 2

Reviewer 1 Report

The authors attempt to address reviewer comments. 

Author Response

Respected Reviewer

Authors are very much thankful to you for your suggestions and comments. This helped us in improving the manuscript.

Reviewer 2 Report

1. Porosity data should be checked again.

2. Unit of x-axis in Figure 1 should be added. 

Author Response

Respected reviewer

Authors are grateful for your suggestions and comments. Please find the attached file herewith
